# Trauma-Informed Understanding of Depression Among Justice-Involved Youth

**DOI:** 10.3390/ijerph22091371

**Published:** 2025-08-31

**Authors:** Richard Dembo, Alexis Swezey, Rachel Herrera, Luz Melendez, Camille Geiger, Kerry Bittrich, Jennifer Wareham, James Schmeidler

**Affiliations:** 1Criminology Department, University of South Florida, Tampa, FL 33620, USA; 2ACTS, Inc. (Agency for Community Treatment Services), Tampa, FL 33604, USA; aswezey@actsfl.org (A.S.); rherrera@actsfl.org (R.H.); lmelendez@actsfl.org (L.M.); cgeiger@actsfl.org (C.G.); kbittrich@actsfl.org (K.B.); 3Wayne State University, Detroit, MI 48201, USA; jwareham@wayne.edu; 4Mt. Sinai Medical School, New York, NY 10029, USA; jim.schmeidler@outlook.com

**Keywords:** justice involved youth trauma, depression among youth sexual victimization, depression

## Abstract

The association between adverse childhood experiences (ACEs) and depression has been the focus of a number of prevalent studies in recent years—particularly among high-risk youth. Depression remains a significant mental health issue among justice-involved youth. There is a well-established correlation between depressed mood and conduct problems (e.g., conduct disorder and oppositional defiant disorder) during childhood and adolescence, which tends to become more prevalent during adolescence. Studies of justice-involved youth reveal high prevalence rates of depression and other mood disorders. Drawing on the relevant literature, we conducted multigroup structural equation model (SEM) analyses to assess the relationships between experiencing ACEs, sexual assault victimization, and depression among male (*n* = 226) and female (*n* = 98) youth entering a post-arrest intake facility in the Florida, U.S.A. juvenile justice system in 2024–2025. The youths averaged 15 years in age, and most were attending middle school or high school. Confirmatory factor analyses (CFAs) were completed to estimate a latent variable labeled depression. Sexual assault victimization and ACEs were hypothesized to be related to each other and were specified as predictors of depression. This trauma/stress experiences and depression model was estimated in two multigroup analyses, across birth gender groups (male or female) and race groups (non-Black or Black) for the youth in this study. The results indicated that there are several notable conclusions from the SEM analyses. First, depression was a scalar invariant in the two multigroup analyses, permitting clearer comparisons of the specified predictors of this construct across groups. Second, for the race-based SEM, experience of sexual assault and the total ACE score were significantly related only in the model for Black youth. The fit of the model was “poorest” among non-Black youth, although even in this case, sexual assault experiences were a significant predictor of depression. Finally, for the gender-based model, sexual assault and ACEs were significant predictors of depression among both male and female youth. Model fit results underscore the important role of abuse trauma and ACEs in understanding these youths’ depression symptoms, and they help contribute to the literature on this topic.

## 1. Introduction

Early exposure to adverse and traumatic situations potentially contributes to the onset of psychological and behavioral problems across developmental stages [1,2,3]. Childhood adversities include physical, psychological, and sexual abuse, neglect, witnessing abuse and violence, and exposure to other stressful events, such as death, separation, or incarceration of a loved one. Adverse childhood experiences are also risk factors for various mental health problems, including PTSD, anxiety, and body image disorders [4].

Exposure to trauma and adverse situations are also associated with changes in the structure of the brain that affect behavior and psychopathology [4,5,6,7,8,9]. From a criminological perspective, theories like general strain theory [10] postulate that stressful situations may motivate deviant coping strategies and negative emotionality. While the direct connection between childhood exposure to stress/trauma and mood disorders remains uncertain, the links between childhood adversities and sexual assault and adolescent mood dysfunction may be important to inform treatment and prevention efforts.

The association between adverse childhood experiences (ACEs) and depression has been a focus of a number of prevalent studies. For example, Elmore and Crouch [11] analyzed cross sectional data from the nationally representative sample of children and adolescents aged 8 to 17 years who participated in the 2006–2017 National Survey of Children’s Health to examine the relationship between ACEs and current anxiety or depression. Using caregiver reports of ACE exposures and current depression, the authors found that 9% of children had current anxiety while 4% had current depression. Multivariate analysis determined that nine of their ACE measures were associated with significantly higher odds of both anxiety and depression, with ACE exposure being more strongly related to depression. Xiao, Zhao, and Zhou [12] conducted a review of the literature on the effects of childhood trauma on adolescent mental health functioning. Their systematic search identified articles on the impact of childhood trauma, such as emotional abuse, physical abuse, and/or neglect, on later depression as a topic of increasing interest in the literature in recent years. They estimated the rate of depression among youth experiencing early aged trauma is 3.79 times greater than among children without such a history.

Research has been mixed and limited on gender and racial differences in the association between ACEs and depression. In a meta-analysis of gender differences in ACEs and mental disorders in 68 studies, Sahle et al. [13] found that ACEs increased the odds of depression, but this effect was not significantly different across gender. In another meta-analysis of studies examining gender differences in the association between ACEs and mental health, Zhu et al. [14] found there were no overall significant gender differences between the number of ACEs and depression; however, there were minor gender differences in the types of ACE and depression. Studies of racial effects on the ACE–depression connection found that white adolescents with more ACEs were at a greater risk of depression compared to Black youth [15,16].

Youths involved in the juvenile justice system are at an even greater risk of adverse childhood experiences than youths within the general population. Wolff and Baglivio [17] reported that justice-involved youths experienced four times as many ACEs as other youths. In a review of studies of the prevalence of ACEs, Malvaso et al. [18] found that the odds of experiencing at least one ACE were 12 times higher for justice-involved youth than non-justice-involved youths. Furthermore, in their meta-analysis, Madigan et al. [19] found the prevalence of four or more ACEs was higher among justice-involved youths compared to non-justice-involved youths. An association between ACEs and depression has also been found among samples of justice-involved adolescents [20].

Relatively little research has explored gender and race differences in adverse childhood experiences among justice-involved adolescents, although we know such experiences often result in significant adverse mental health outcomes [20]. Adult justice-involved samples tend to find females report higher rates of childhood physical, psychological/emotional, and sexual abuse than males [21,22,23]. The present study contributes to the scant literature on ACEs and depression among justice-involved youth.

Mental health risk concerns of justice-involved youth often include their levels of depression, which tend to relate to other co-occurring mental health problems, such as post-traumatic stress disorder (PTSD), trauma, and substance misuse, and to long-term issues of depression and associated negative outcomes [24,25,26]. Depressed mood often remains a significant mental health problem among justice-involved youth [27]. There is a well-established correlation between depressed mood and conduct problems (e.g., conduct disorder and oppositional defiant disorder) during childhood and adolescence [28,29], which tends to become more prevalent during adolescence [30]. Studies of youth involved in delinquency who enter the justice system have revealed high prevalence rates of depression and other mood disorders [31,32]. For example, Teplin, Abram, McClelland, Dulcan, and Mericle [33] found that two-thirds of Cook County detained males and almost three-quarters of detained females met diagnostic criteria for one or more psychiatric disorders. Half of the males and females had a substance use disorder. Affective disorders were also prevalent, especially among females, with 20% meeting diagnostic criteria for a major depressive episode [33]. Other studies have identified similar prevalence rates of depressive symptoms and substance use disorders [34,35]. Related research on justice-involved youth has also identified an association between depression and sexual risk behaviors [36], and comorbidity in marijuana use and depression [31,37]. Moreover, longitudinal studies suggest early problems with both depression and delinquency can lead to difficulties with school success, substance use, continued depression, and criminal offending in adulthood [29,38,39,40].

In an important longitudinal study, Teplin and her associates [26] found that, among Illinois detained youth in Cook County, 52.3% of males and 30.9% of females still had a psychiatric disorder 15 years after detention. The prevalence of psychiatric disorders was higher than in the general population; males were more likely to persist with a psychiatric disorder than females, and untreated traumas and other disorders made it harder to finish school, get a job, and stay out of jail. The study also found that males with depressive disorder at baseline were more likely than those without this disorder to have mood, anxiety, and alcohol use problems 15 years later. Females with depressive disorders at baseline were more likely than those without to have nearly all disorders [26]. Johnson and colleagues note that while research has improved our knowledge of “disruptive behavior disorders among incarcerated youths, less is known about the factors associated with depression and other internalizing symptoms in this population” [41] (p. 1096). Research on gender and racial/ethnic differences in depression among justice-involved samples is limited and mixed. For example, Wareham et al. [42] studied depression and childhood sexual assault among justice-involved youth and found that white, male youth had greater odds of longitudinal depression symptoms compared to other youths. In a meta-analysis of 30 studies of samples of justice-involved youth, Livanou et al. [43] found that females had higher prevalence rates of depression, compared to males. Other research demonstrated no significant gender or race/ethnicity differences in the prevalence of depression among justice-involved youths [34]. The present study contributes to the literature examining gender and race/ethnicity differences in depression among justice-involved youth.

Childhood sexual assault is associated with a variety of negative correlates and consequences. Longitudinal studies over the life course find that children and adolescents are at higher risk of victimization [44,45,46], particularly from a family member [45]. Depression, in particular, is a more common consequence of childhood sexual abuse [47,48,49]. Among justice-involved youth, studies find comorbidity of depression and sexual assault victimization [50,51].

Gender differences exist in the prevalence of childhood sexual assault victimization, with girls generally demonstrating higher rates than boys [52,53,54]. Youth involved in the juvenile justice system are at a greater risk of victimization than non-justice-involved youth [51,55,56]. With regard to gender differences in the negative consequences of childhood sexual assault victimization, the results have been mixed. Some studies find gender differences, such as girls demonstrating more internalizing symptoms than boys [48,57] and boys demonstrating more negative consequences than girls overall [58]. Yet other studies find no gender differences in consequences of childhood sexual assault [59,60,61].

Research on race/ethnicity differences in childhood sexual assault has been limited, especially among justice-involved youth. These studies have reported mixed findings with some finding racial/ethnic differences and others none [62,63,64,65]. In a study of childhood sexual assault among justice-involved youth, Dembo et al. [66] found that white, female youth reported higher rates of sexual assault than their male or Black or Hispanic female counterparts. This study contributes to the literature examining gender and race/ethnicity differences in childhood sexual assault among justice-involved youth.

## 2. Estimated Model

Drawing on the above discussed literature, we formulated and estimated a multigroup structural equation model (SEM) to test several hypotheses illustrated in Figure 1. First, we anticipated the SEM latent construct for depression with observed covariates for age, ACEs, and sexual assault would fit the data in the study. Confirmatory factor analyses (CFAs) were conducted on an eight-item depression measure to estimate a latent variable labeled depression. Second, self-reports of sexually assault victimization and ACEs were hypothesized to be positively related to each other and positively associated with the latent variable of depression. Further, age was hypothesized as a predictor of depression. This trauma/stress experiences and depression model for youth was estimated in two multigroup SEMs: birth gender groups (male and female) and race/ethnicity groups (non-Black and Black). Since the literature on gender- and race-based differences in the model is limited and mixed, we did not specify how the above hypotheses would vary across the groups.

## 3. Methodology

### 3.1. Procedure

The data for this study were collected as part of an innovative health coach service, a collaboration between a community-based behavioral health organization (ACTS, Inc.), the county health department, and a local juvenile justice system facility. As described in earlier studies [67], data were collected in a health service for youths entering a juvenile assessment center (JAC), a centralized intake facility located in a southeastern U.S. city. The collaborative service had three major goals:To offer HIV and sexually transmitted infection (STI) evidence-based risk reduction information and education to youths using gender- and developmentally appropriate curricula.To perform analysis of youths’ provided urine specimens to test for drug use and STIs and swab testing for HIV.To connect youth with identified needs to appropriate community services. The service sought to follow-up with STI- and HIV-positive youth and promptly link them with appropriate treatment. Youth who screened high on an evidence-based depression inventory [68,69] were also linked with follow-up services.

Health coaches possessed at least an undergraduate degree and were trained to follow a detailed data collection and service delivery protocol, including Department of Health’s STI and HIV pretest and post-test counseling. Health coaches were selected based on their abilities to interact effectively with adolescents and discuss sensitive sexual behavior, substance use, and mental health issues. Health coaches were trained to inform youth entering the JAC subsequent to criminal charges or arrests of the purpose of the health coach project, the types of information and biological assays youths would be asked to provide, and how this information would be protected and utilized. If needed, youths were referred to other follow-up health-related services. Youths were not compensated for participation. Data collection and entry were routinely monitored for integrity and quality by the program manager (a coauthor of this article). Data were collected in accordance with the requirements of the Institutional Review Board. Data were de-identified and shared electronically with the senior researcher for this study.

### 3.2. Participants

Participation in the health coach services for the present data occurred from 3 September 2024 to 3 February 2025. Following informed consent, *n* = 98 girls and *n* = 226 boys received health coach services. Twenty-five (7.7%) of the total *n* = 324 justice-involved youths entered the JAC more than once, during which occasion(s) they received additional health coach services. For youth with multiple entries, only health coach data collected during their first entry were used in the present study. Sociodemographic information (gender, age, race/ethnicity) was also available for all youth. Agreeing to receive this service was voluntary. Florida public health law does not require youths 12 years and over to obtain parental consent for STI or HIV testing or treatment. Overall, the participation rate in the health coach services was 71.9%. No routinely available data were collected on youth who declined to participate in health coach services.

### 3.3. Measure of Depression

We used the eight-item version of the widely used twenty-item Center for Epidemiological Studies Depression Scale (CES-D) [68,69] derived from the psychometric work of Melchior and colleagues [70] to measure depression (also see: [71,72]). The following are the items used: (1) “I felt I could not shake off the blues even with the help from my family and friends”; (2) “I felt sad”; (3) “I felt depressed”; (4) “I thought my life had been a failure”; (5) “I felt fearful”; (6) “My sleep was restless”; (7) “I felt lonely”; and (8) “I had crying spells.” The time frame used for these experiences was the past week. Each item was scored as follows: (0) less than one day, (1) 1 to 2 days, (2) 3 to 4 days, and (3) 5 to 7 days. When used among justice-involved youth, the 8-item depression measure has demonstrated excellent psychometric properties [27]. Cronbach’s alpha for the depression items in this study was 0.83.

### 3.4. Covariate Measures

Several demographic data were collected, including (a) age (in number of years); (b) birth gender (male = 0, female = 1); and (c) information on the youths’ race/ethnicity (African American or Black = 1, Hispanic = 2, Anglo American = 3, and other [one Asian youth received health coach services] = 4). Since only 45 Hispanic youth (16 females, 29 males) received health coach services, we were unable to conduct fruitful multigroup structural equation analyses for this group. We plan to conduct such analyses in the future.

We collected information on the youths’ ACEs [73]. The ACE questions (details about the instrument are included in Appendix A) asked about the following childhood experiences (percent reporting each experience noted in parentheses): (1) physical abuse (4.5%); (2) sexual abuse (i.e., fondled or touched in a sexual way or attempted/performed sexual act; 2.1%); (3) emotional abuse (6.5%); (4) physical neglect (1.0%); (5) emotional neglect (5.5%); (6) mental illness (3.4%); (7) incarcerated relative (16.8%); (8) mother treated violently (5.5%); (9) substance use (6.2%); and (10) divorce (72.9%). Consistent with studies utilizing the ACEs, each ACE item was a dichotomous indicator (coded 0 = no, 1 = yes). Each youth received a total ACE score that was used in the analyses.

As indicated in the Appendix A for item 2, the ACE item for sexual abuse referred to fondling, touching, and sexual acts attempted or committed by an adult or person at least five years older than the study participant. Since this item does not specify that the act occurred without the youth’s consent, we also asked each youth if he or she had ever been sexually assaulted. Sexual assault requires a lack of consent and includes a broader range of acts than the single ACE item. While there is some conceptual overlap between these items, the correlation between the items was low (*r* = 0.28, *p* < 0.001). Based on this weak correlation, it was unlikely there were multicollinearity issues introduced by including both total ACEs and sexual assault in the multigroup SEM analyses. Responses were coded (0) for no reports of being sexually assaulted and (1) for experiencing sexual assault.

### 3.5. Strategy of Analysis

Descriptive data were analyzed using SPSS version 29 [74]. The multigroup structural equation analyses were conducted using Mplus version 8.11 [75].

## 4. Results

### 4.1. Sample Description

Descriptive statistics for age, birth gender, race, ACEs, sexual assault victimization, and the depression items are reported in Table 1. There were more boys (69.8% to 30.2%) than girls in this study. A majority of the males and females self-identified as African American or Black (73.0% and 65.3%, respectively). The males and females averaged 15.5 and 15.0 years of age, respectively. The females (*M* = 1.44) reported significantly higher total ACEs scores than the males (*M* = 1.01). A significantly larger percent of females (16.8%) than males (0.9%) reported sexual assault.

In regard to depression, females reported significantly higher levels on four of the eight depression items than the males (felt sad, felt depressed, felt fearful, and had crying spells). Importantly, 22.1% of girls had a depression total score of 7 or higher, a designated threshold score indicative of potentially needing clinical intervention [76,77], compared to 12.8% for the boys.

In regard to race, the majority of youth were male (Black = 72.1%, non-Black = 64.2%), near in age (Black = 15.4 years, non-Black = 15.2 years), and with similar ACE average scores (Black = 1.13, non-Black = 1.17). These youth had similar scores on the individual depression items, with one exception. Non-Black youth reported a significantly higher rate of thinking their life was a failure (0.45),than Black youth (0.26). More non-Black youth (18.5%) had a depression score of 7 or higher than Black youth (14.4%). In regard to experiencing sexual assault, significantly more non-Black youth (10.9%) than Black youth (3.6%) reported this.

### 4.2. Trauma/Stress Experiences and Depression

The multigroup models were estimated in Mplus 8.11 by MLR, maximum likelihood of parameter estimates with standard errors and a chi-square statistic (when applicable) that are robust to non-normality and non-independence of observations [75]. Prior to estimating the structural equation model specified in Figure 1, we conducted invariance analyses [78] of the depression measure across the male and female and non-Black and Black youth groups. The model analyses confirmed metric invariance for the CFA across the male and female youth groups (one factor and equal factor loadings), and scalar invariance across the non-Black and Black youth groups (one factor, equal factor loadings, and equal thresholds). (Detailed statistical analysis results available from the senior author upon request.)

### 4.3. Model Results for Males and Females

For each gender group, model estimation results indicated significant, positive relationships between experiencing sexual assault and total reported ACEs and level of depression. An excellent model fit was found. Examination of the univariate and bivariate distributions of standardized residuals found relatively few significant residuals. Table 2 presents statistical details of the analysis results. For both boys and girls, the association between ACEs and sexual assault (males: estimate/SE = 0.936, *p* = 0.349; females: estimate/SE = 1.902, *p* = 0.057) and between age and depression (males: estimate/SE = −1.371, *p* = 0.170; females: estimate/SE = 1.119, *p* = 0.263) was non-significant. For both boys and girls, there was a significant relationship between ACEs and depression (males: estimate/SE = 3.158, *p* = 0.002; females: estimate/SE = 2.212, *p* = 0.027) and sexual assault and depression (males: estimate/SE = 3.609, *p* < 0.001; females: estimate/SE = 3.042, *p* = 0.002).

### 4.4. Model Results for Non-Black and Black Youths

Table 3 presents the results of the estimation of the trauma/stress experience–depression model for non-Black and Black youth. Among non-Black youth, depression was significantly predicted by experience(s) of sexual assault. Among Black youth, both experience of sexual assault and total ACEs were related to each other, and each significantly predicted depression. Again, an excellent model fit was found. Examination of the univariate and bivariate distributions of standardized residuals found relatively few significant residuals. Table 3 provides statistical details on the analysis results. The association between ACEs and sexual assault was non-significant for non-Black youth (estimate/SE = 2.548, *p* = 0.122) but significant for Black youths (estimate/SE = 1.976, *p* = 0.048). For both non-Black and Black youths, there was a non-significant relationship between age and depression (non-Black: estimate/SE = −0.181, *p* = 0.856; Black: estimate/SE = 0.033, *p* = 0.973). For both non-Black and Black youths, sexual assault was significantly related to depression (non-Black: estimate/SE = 2.714, *p* = 0.007; Black: estimate/SE = 2.910, *p* = 0.004). However, the total ACE score was only significantly related to depression for Black youths (estimate/SE = 3.438, *p* = 0.001), not non-Black youths (estimate/SE = 1.131, *p* = 0.258).

## 5. Discussion

This study examined gender- and race/ethnicity-based differences in the association between childhood adversities and depressive symptoms among a sample of justice-involved adolescents. Findings indicate the latent structure for depression (see Figure 1) was invariant, supporting metric invariance for boys vs. girls and scalar invariance for Black youths vs. non-Black youths. Finding invariance in the two multigroup analyses indicates comparisons of the factor means with the predictors are appropriate across groups. Consequently, these results support the utility of the Center for Epidemiological Studies Depression Scale as an instrument that is generalizable regarding measurement to different demographic groups. Since the CES-D is a brief instrument that is easily administered to youth, this depression assessment is a good option for juvenile justice system administrators and related agencies for use during intake procedures, which must often meet strict time constraints for completion. Overall, the results for the depression factor structure support our expectation that the model would fit the data.

The findings regarding the hypothesized positive association between ACEs and sexual assault and depression are mixed. Contrary to our hypothesis, sexual assault was only significantly correlated with ACEs for Black youths, not non-Black youths or girls and boys (regardless of race/ethnicity). This finding is interesting because the literature on childhood adversity suggests a connection between ACEs and sexual assault victimization [44,45,46] and one item in the total ACEs index refers to a form of sexual assault. On the one hand, this finding suggests there is little measurement overlap between the ACEs score and the separate indicator for sexual assault used in our study. On the other hand, this finding suggests controlling for depression and age may reduce the association between ACEs and sexual assault victimization. Future research is needed to explore this effect in more detail.

We found support for the hypothesized positive relationship between sexual assault victimization and depression. For each subgroup, sexual assault victimization increased the odds of a youth reporting higher levels of depressive symptoms. These results are consistent with prior research that has noted depression as a consequence of sexual assault victimization among justice-involve youth [50,51]. It may benefit practitioners in the juvenile justice system to include a brief questionnaire on sexual assault (and other victimization) as part of the intake process for youths.

We found mixed results regarding the ACE-depression connection, with the effect being positive and significant for boys, girls, and Black youths but non-significant for non-Black (primarily white) youths. Although this finding is contrary to our hypothesis, it is consistent with prior research that found no gender effects [13,14] for the association between ACEs and depression and an increased risk of depression for white youths with higher ACEs but not Black youths [15,16]. While inclusion of an ACE survey may benefit practitioners in juvenile justice to identify at-risk youth, additional research is needed to determine if the findings here are replicable and whether this relationship is spurious.

Furthermore, our results indicated no significant association between age and depression. We did not hypothesize a direction for the relationship between age and depression. Controlling for the influence of ACEs and sexual assault victimization, age seemed less important for predicting depression among the justice-involved youths in this study.

## 6. Study Limitations and Their Implications

There are several limitations to the research reported in this paper. First, the multigroup, multivariate analyses were conducted on cross-sectional data. Hence, no causal interpretations of our findings are possible. Second, we did not have a sufficient number of Hispanic youths to perform meaningful psychometric and SEM analyses for them. Relatedly, the relatively small number of cases we studied reduced statistical power and prevented completion of other informative multivariate analyses, such as latent class analyses. Such studies are planned for the future. Third, our sample of youth was geographically limited. Hence, the results of this study may not generalize to male and female youths arrested in other jurisdictions, reflecting different sociodemographic and contextual circumstances. Future research should conduct similar studies in other jurisdictions. Fourth, this study relied on a total score for ACEs and did not examine differences for youths who experienced a certain threshold of adversity (such as 4 or more of the 10 ACE areas) or certain types of ACEs. Research has suggested justice-involved youth are especially at risk of experiencing a higher number of ACEs [17,18,19]. Moreover, studies have also suggested differences in negative outcomes based on the type of adversity [79,80]. Future research should explore how variations in number and type of ACEs affect psychopathology among justice-involved youth. Fifth, this study relied on adverse childhood experiences as a covariate of depression. More recently, studies of positive compared to adverse childhood experiences and mental health issues among youth have demonstrated the protective effects of positive childhood experiences may be more impactful than the deleterious effects of ACEs [81,82].

Future studies should explore the impact of positive versus adverse childhood experiences on mental health among justice-involved youth. In this vein, future research should follow a service system cascade model involving the identification of service need followed by indicated intervention/treatment engagement within a longitudinal framework to address the long term psychosocial and behavioral manifestations of ACEs and trauma abuse experiences [27].

## 7. Conclusions

This study contributed to the scant research on childhood adversity and depression among justice-involved youth. Overall, sexual assault and ACEs were associated with depression for girls, boys, and Black youths, while sexual assault and not ACEs were associate with depression for non-Black youths. These findings suggest childhood adversities are comorbid with depressive symptoms among justice-involved youths.

## 8. Implications of Our Research for Practice

The results of our research strongly recommend that youth entering the justice system be routinely screened for childhood adversity, abuse experiences, and mental health issues in a manner that elicits honest reporting of these experiences, with particular reference to gender and race/ethnicity differences in these phenomena. It is, of course, critical that these screenings be followed, where indicated, by full assessments and treatment follow-up. Unfortunately, such a cascade of treatment services is not routinely available in many juvenile justice settings.

There is a continuing need to inform policy makers on the public health issues and service needs among justice-involved youth, such as the youth we studied. Providing preventive and early intervention mental health and related services (e.g., drug use problems) to these at-risk youth holds great promise for reducing their flow into the juvenile, and subsequent adult, justice systems with increased chances of future conflict with the law experiences. Continued advocacy is needed to address this pressing public health policy crisis.

## Figures and Tables

**Figure 1 ijerph-22-01371-f001:**
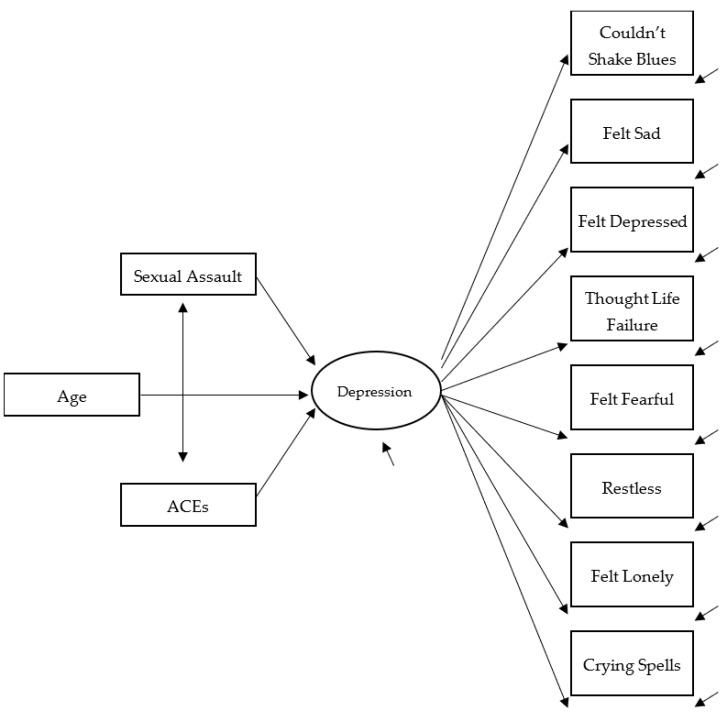
Trauma/Stress experiences and depression. Structural equation model illustrating a latent variable for depression comprising eight observed items, covariates of age, adverse childhood experiences (ACEs), and sexual assault victimization related to depression, and correlation between ACEs and sexual assault victimization.

**Table 1 ijerph-22-01371-t001:** Demographic characteristics and description of major variables.

	Male	Female	Non-Black	Black
	(*n* = 219–226)	(*n* = 95–98)	(*n* = 92–95)	(*n* = 222–229)
Male	---	---	64.2%	72.1%
Female	---	---	35.8%	27.9%
			100.0%	100.0%
Age, mean (*SD*)	15.50 (1.31)	15.01 (1.54) **	15.15 (1.52)	15.43 (1.34)
ACEs total	1.01 (0.99)	1.44 (1.50) **	1.17 (1.37)	1.13 (1.10)
Sexually assaulted	0.9%	16.8% **	10.9%	3.6% *
Depression items:				
Could not shake off blues	0.15 (0.56)	0.22 (0.66)	0.22 (0.63)	0.15 (0.57)
Felt sad	0.44 (0.86)	0.76 (1.02) **	0.67 (1.02)	0.48 (0.87)
Felt depressed	0.29 (0.80)	0.60 (1.05) **	0.47 (0.97)	0.35 (0.86)
Thought my life a failure	0.26 (0.73)	0.43 (0.81)	0.45 (0.86)	0.26 (0.71) *
Felt fearful	0.14 (0.48)	0.33 (0.76) **	0.22 (0.57)	0.19 (0.59)
Sleep was restless	0.66 (1.12)	0.67 (1.07)	0.74 (1.17)	0.63 (1.08)
Felt lonely	0.34 (0.84)	0.52 (1.05)	0.50 (0.99)	0.35 (0.88)
Had crying spells	0.14 (0.54)	0.55 (0.91) ***	0.35 (0.73)	0.23 (0.68)

Note. ACEs = adverse childhood experiences. Two-tailed *p*-values: * *p* < 0.05; ** *p* < 0.01; *** *p* < 0.001.

**Table 2 ijerph-22-01371-t002:** Gender-based multigroup SEM results: males (*n* = 220) vs. females (*n* = 96).

Model Fit Information (Unstandardized Estimates)				
Number of free parameters				48
Loglikelihood:				
H0 value				−2043.929
H0 scaling correction factor for MLR				1.284
Information criteria:				
Akaike (AIC)				4183.858
Bayesian (BIC)				4364.133
Sample-size adjusted BIC				4211.889
**Final class counts and proportions for the latent classes based on the estimated model**				
Latent classes:	Count	Proportion		
1 (0 = Male)	220	0.696		
2 (1 = Female)	96	0.304		
**Model results by latent class (gender group)**				
	Estimate	SE	Estimate/SE	*p*-value
Latent class 1: Male				
Depression by:				
Could not shake off blues	1.000	0.000	---	---
Felt sad	1.231	0.288	4.277	**0.000**
Felt depressed	1.402	0.409	3.427	**0.001**
Thought my life a failure	0.954	0.222	4.307	**0.000**
Felt fearful	0.847	0.190	4.457	**0.000**
Sleep was restless	0.584	0.137	4.259	**0.000**
Felt lonely	1.207	0.299	4.044	**0.000**
Had crying spells	0.987	0.253	3.909	**0.000**
Depression on:				
ACEs	0.654	0.207	3.158	**0.002**
Sexual assault	2.796	0.774	3.609	**0.000**
Age	−0.173	0.126	−1.371	0.170
ACEs with:				
Sexual assault	0.102	0.109	0.936	0.349
Means:				
ACEs	1.009	0.067	15.106	**0.000**
Sexual assault	0.009	0.007	1.329	0.184
Intercepts:				
Depression	4.866	3.542	1.374	0.170
Thresholds:				
Could not shake off blues $1	6.920	3.067	2.256	**0.024**
Could not shake off blues $2	7.417	3.082	2.407	**0.016**
Could not shake off blues $3	9.042	3.187	2.837	**0.005**
Felt sad $1	5.484	3.248	1.688	0.091
Felt sad $2	7.012	3.281	2.137	**0.033**
Felt sad $3	8.747	3.303	2.648	**0.008**
Felt depressed $1	7.758	3.740	2.074	**0.038**
Felt depressed $2	8.382	3.762	2.228	**0.026**
Felt depressed $3	9.762	3.815	2.559	**0.010**
Thought my life a failure $1	5.513	2.535	2.175	**0.030**
Thought my life a failure $2	6.601	2.546	2.593	**0.010**
Thought my life a failure $3	7.801	2.617	2.981	**0.003**
Felt fearful $1	5.615	2.415	2.324	**0.020**
Felt fearful $2	6.793	2.421	2.806	**0.005**
Felt fearful $3	8.093	2.452	3.301	**0.001**
Sleep was restless $1	2.920	1.605	1.819	0.069
Sleep was restless $2	3.444	1.613	2.135	**0.033**
Sleep was restless $3	4.187	1.619	2.585	**0.010**
Felt lonely $1	6.765	3.191	2.120	**0.034**
Felt lonely $2	7.276	3.196	2.277	**0.023**
Felt lonely $3	8.316	3.226	2.578	**0.010**
Had crying spells $1	5.935	2.688	2.208	**0.027**
Had crying spells $2	7.021	2.731	2.571	**0.010**
Had crying spells $3	8.343	2.840	2.938	**0.003**
Variances:				
ACEs	1.462	0.352	4.150	**0.000**
Sexual assault	0.050	0.010	5.114	**0.000**
Residual variances:				
Depression	3.558	1.305	2.726	**0.006**
Latent class 2: Female				
Depression by:				
Could not shake off blues	1.000	0.000	---	---
Felt sad	1.231	0.288	4.277	**0.000**
Felt depressed	1.402	0.409	3.427	**0.001**
Thought my life a failure	0.954	0.222	4.307	**0.000**
Felt fearful	0.847	0.190	4.457	**0.000**
Sleep was restless	0.584	0.137	4.259	**0.000**
Felt lonely	1.207	0.299	4.044	**0.000**
Had crying spells	0.987	0.253	3.909	**0.000**
Depression on:				
ACEs	0.388	0.175	2.212	**0.027**
Sexual assault	2.287	0.752	3.042	**0.002**
Age	0.201	0.180	1.119	0.263
ACEs with:				
Sexual assault	0.035	0.018	1.902	0.057
Means:				
ACEs	1.438	0.152	9.442	**0.000**
Sexual assault	0.168	0.038	4.386	0.000
Intercepts:				
Depression	0.000	0.000	---	---
Thresholds:				
Could not shake off blues $1	6.920	3.067	2.256	**0.024**
Could not shake off blues $2	7.417	3.082	2.407	**0.016**
Could not shake off blues $3	9.042	3.187	2.837	**0.005**
Felt sad $1	5.484	3.248	1.688	0.091
Felt sad $2	7.012	3.281	2.137	**0.033**
Felt sad $3	8.747	3.303	2.648	**0.008**
Felt depressed $1	7.758	3.740	2.074	**0.038**
Felt depressed $2	8.382	3.762	2.228	**0.026**
Felt depressed $3	9.762	3.815	2.559	**0.010**
Thought my life a failure $1	5.513	2.535	2.175	**0.030**
Thought my life a failure $2	6.601	2.546	2.593	**0.010**
Thought my life a failure $3	7.801	2.617	2.981	**0.003**
Felt fearful $1	5.615	2.415	2.324	**0.020**
Felt fearful $2	6.793	2.421	2.806	**0.005**
Felt fearful $3	8.093	2.452	3.301	**0.001**
Sleep was restless $1	2.920	1.605	1.819	0.069
Sleep was restless $2	3.444	1.613	2.135	**0.033**
Sleep was restless $3	4.187	1.619	2.585	**0.010**
Felt lonely $1	6.765	3.191	2.120	**0.034**
Felt lonely $2	7.276	3.196	2.277	**0.023**
Felt lonely $3	8.316	3.226	2.578	**0.010**
Had crying spells $1	5.935	2.688	2.208	**0.027**
Had crying spells $2	7.021	2.731	2.571	**0.010**
Had crying spells $3	8.343	2.840	2.938	**0.003**
Variances:				
ACEs	1.462	0.352	4.150	**0.000**
Sexual assault	0.050	0.010	5.114	**0.000**
Residual variances:				
Depression	3.558	1.305	2.726	**0.006**
Categorical latent variables:				
Means:				
G #1 (males)	0.829	0.122	6.780	**0.000**

Note. ACEs = adverse childhood experiences. Significant *p*-values < 0.05 in bold.

**Table 3 ijerph-22-01371-t003:** Race-based multigroup SEM results: non-Black (*n* = 93) vs. Black (*n* = 223).

Model Fit Information (Unstandardized Estimates)				
Number of free parameters				48
Loglikelihood:				
H0 value				−2055.514
H0 scaling correction factor for MLR				1.267
Information criteria:				
Akaike (AIC)				4207.028
Bayesian (BIC)				4387.304
Sample-size adjusted BIC				4235.060
**Final class counts and proportions for the latent classes based on the estimated model**				
Latent classes:	Count	Proportion		
1 (0 = non-Black)	93	0.294		
2 (1 = Black)	223	0.706		
**Model results by latent class (race group)**				
	Estimate	SE	Estimate/SE	*p*-value
Latent class 1: Non-Black				
Depression by:				
Could not shake off blues	1.000	0.000	---	---
Felt sad	1.176	0.267	4.397	**0.000**
Felt depressed	1.312	0.380	3.458	**0.001**
Thought my life a failure	0.922	0.211	4.361	**0.000**
Felt fearful	0.805	0.181	4.442	**0.000**
Sleep was restless	0.569	0.130	4.389	**0.000**
Felt lonely	1.185	0.293	4.044	**0.000**
Had crying spells	0.938	0.231	4.057	**0.000**
Depression on:				
ACEs	0.276	0.244	1.131	0.258
Sexual assault	1.766	0.651	2.714	**0.007**
Age	−0.030	0.167	−0.181	0.856
ACEs with:				
Sexual assault	0.026	0.017	1.548	0.122
Means:				
ACEs	1.172	0.142	8.281	**0.000**
Sexual assault	0.108	0.032	3.347	**0.001**
Intercepts:				
Depression	1.713	3.368	0.509	0.611
Thresholds:				
Could not shake off blues $1	5.034	2.370	2.124	0.034
Could not shake off blues $2	5.539	2.360	2.348	0.019
Could not shake off blues $3	7.192	2.420	2.972	0.003
Felt sad $1	2.964	2.545	1.164	0.244
Felt sad $2	4.485	2.591	1.731	0.083
Felt sad $3	6.219	2.645	2.351	**0.019**
Felt depressed $1	4.807	2.796	1.719	0.086
Felt depressed $2	5.418	2.815	1.925	0.054
Felt depressed $3	6.774	2.895	2.340	**0.019**
Thought my life a failure $1	3.591	1.992	1.803	0.071
Thought my life a failure $2	4.684	1.997	2.345	**0.019**
Thought my life a failure $3	5.889	2.096	2.810	**0.005**
Felt fearful $1	3.869	1.739	2.225	**0.026**
Felt fearful $2	5.042	1.722	2.928	**0.003**
Felt fearful $3	6.339	1.732	3.660	**0.000**
Sleep was restless $1	1.746	1.225	1.425	0.154
Sleep was restless $2	2.274	1.226	1.855	0.064
Sleep was restless $3	3.022	1.244	2.429	**0.015**
Felt lonely $1	4.394	2.503	1.756	0.079
Felt lonely $2	4.914	2.510	1.958	0.050
Felt lonely $3	5.970	2.542	2.348	**0.019**
Had crying spells $1	3.894	2.542	2.348	**0.019**
Had crying spells $2	4.974	2.076	2.396	**0.017**
Had crying spells $3	6.293	2.173	2.895	**0.004**
Variances:				
ACEs	1.425	0.230	6.196	**0.000**
Sexual assault	0.053	0.011	4.722	**0.000**
Residual variances:				
Depression	3.926	1.425	2.755	**0.006**
Latent class 2: Black				
Depression by:				
Could not shake off blues	1.000	0.000	---	---
Felt sad	1.176	0.267	4.397	**0.000**
Felt depressed	1.312	0.380	3.458	**0.001**
Thought my life a failure	0.922	0.211	4.361	**0.000**
Felt fearful	0.805	0.181	4.442	**0.000**
Sleep was restless	0.569	0.130	4.389	**0.000**
Felt lonely	1.185	0.293	4.044	**0.000**
Had crying spells	0.935	0.231	4.057	**0.000**
Depression on:				
ACEs	0.704	0.205	3.438	**0.001**
Sexual assault	3.053	1.049	2.910	**0.004**
Age	0.004	0.132	0.033	0.973
ACEs with:				
Sexual assault	0.083	0.042	1.976	**0.048**
Means:				
ACEs	1.126	0.073	15.318	**0.000**
Sexual assault	0.036	0.013	2.880	**0.004**
Intercepts:				
Depression	0.000	0.000	---	---
Thresholds:				
Could not shake off blues $1	5.034	2.370	2.124	**0.034**
Could not shake off blues $2	5.539	2.360	2.348	**0.019**
Could not shake off blues $3	7.192	2.420	2.972	**0.003**
Felt sad $1	2.964	2.545	1.164	0.244
Felt sad $2	4.485	2.591	1.731	0.083
Felt sad $3	6.219	2.645	2.351	**0.019**
Felt depressed $1	4.807	2.796	1.719	0.086
Felt depressed $2	5.418	2.815	1.925	0.054
Felt depressed $3	6.774	2.895	2.340	**0.019**
Thought my life a failure $1	3.591	1.992	1.803	0.071
Thought my life a failure $2	4.684	1.997	2.345	**0.019**
Thought my life a failure $3	5.889	2.096	2.810	**0.005**
Felt fearful $1	3.869	1.739	2.225	**0.026**
Felt fearful $2	5.042	1.722	2.928	**0.003**
Felt fearful $3	6.339	1.732	3.660	**0.000**
Sleep was restless $1	1.746	1.225	1.425	0.154
Sleep was restless $2	2.274	1.226	1.855	0.064
Sleep was restless $3	3.022	1.244	2.429	**0.015**
Felt lonely $1	4.394	2.503	1.756	0.079
Felt lonely $2	4.914	2.510	1.958	0.050
Felt lonely $3	5.970	2.542	2.348	**0.019**
Had crying spells $1	3.894	2.056	1.894	0.058
Had crying spells $2	4.974	2.076	2.396	**0.017**
Had crying spells $3	6.293	2.173	2.895	**0.004**
Variances:				
ACEs	1.425	0.230	6.196	**0.000**
Sexual assault	0.053	0.011	4.722	**0.000**
Residual variances:				
Depression	3.926	1.425	2.755	**0.006**
Categorical latent variables:				
Means:				
G #1 (non-Black)	−0.875	0.123	−7.085	**0.000**

Note. ACEs = adverse childhood experiences. Significant *p*-values < 0.05 in bold.

## Data Availability

Due to concerns about privacy and legal issues involving the youths in this study, data are not available to the public.

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
