# Peer review of "Trauma-Informed Understanding of Depression Among Justice-Involved Youth"

_ijerph, 2025, doi:10.3390/ijerph22091371_

Round 1

Reviewer 1 Report

Comments and Suggestions for Authors

COMMENT 1: The method section could usefully be expanded to include a ‘Procedure’ section that  outlines more precisely (how, where, and by whom) data were collected. In this regards, what efforts (apart from obtaining approval from an IRB) where made to address ethical issues relating to the possibility of research-induced distress among participants. Further, did the consent provided by participants (line 128) specifically include consent/assent for the data to be used for research purposes?

COMMENT 2:   It is far from clear to me why sexual assault and ACEs were considered separately as predictor variables. This appears to me to be redundant as sexual assault is assessed in the ACE-Q  using a probe that is equivalent to the measure of sexual assault devised by the authors (and thus any associations between these two variables identified in the analysis may simply reflect such repeated measurement). I am open to the possibility that I may be missing something here, but I would like the authors to clarify this issue.

COMMENT 3: It would be helpful if the authors could provide more detailed information regarding the frequency of specific ACEs assessed in the study, as well as an indication of the internal consistency of the CES-D in this study.

COMMENT 4: It would assist readers if the author could extend the discussion section to highlight the extent to which study findings were consistent with, or divergent from, findings obtained in the extant literature on the topic (in this regard the discussion section contains no references at all).

COMMENT 5: It would enhance the value of the paper if the authors more clearly highlighted the implications of the study findings (for theory, practice, and future research) .

Author Response

1) The method section could usefully be expanded to include a ‘Procedure’ section that outlines more precisely (how, where, and by whom) data were collected. In this regards, what efforts (apart from obtaining approval from an IRB) where made to address ethical issues relating to the possibility of research-induced distress among participants. Further, did the consent provided by participants (line 128) specifically include consent/assent for the data to be used for research purposes?

We appreciate the reviewer’s recommendation and have revised the first subsection under the Method section to reflect the Procedure of the study. We have added more details about the procedure for the data collection.

2) It is far from clear to me why sexual assault and ACEs were considered separately as predictor variables. This appears to me to be redundant as sexual assault is assessed in the ACE-Q using a probe that is equivalent to the measure of sexual assault devised by the authors (and thus any associations between these two variables identified in the analysis may simply reflect such repeated measurement). I am open to the possibility that I may be missing something here, but I would like the authors to clarify this issue.

The ACE item about sexual abuse differs from sexual assault because the ACE item’s description of sexual abuse may include “consensual” activities and is not as broad as sexual assault. We have added content to the description of the sexual assault measure and why it was used along with the ACE item on sexual abuse. We have also included information about the correlation between these two measures (r = 0.28). Based on the low correlation, it is unlikely any conceptual overlap creates an issue of multicollinearity in the SEM analyses. We hope the revision satisfies the reviewer and readers.

3) It would be helpful if the authors could provide more detailed information regarding the frequency of specific ACEs assessed in the study, as well as an indication of the internal consistency of the CES-D in this study.

The ACEs questionnaire asked respondents to indicate whether each item ever occurred during their childhood. It did not capture the frequency of occurrences. Potential ACE total scores ranged from 0 to 10. We reported the prevalence rate for each item in the original manuscript under the section describing the measure.  

We have added the Cronbach’s alpha score (0.83) for the CES-D measure in the study in the measurement section.

4) It would assist readers if the author could extend the discussion section to highlight the extent to which study findings were consistent with, or divergent from, findings obtained in the extant literature on the topic (in this regard the discussion section contains no references at all).

We have revised the Discussion section to elaborate on the findings and implications and highlight how the results compare to other studies. References are now included in the Discussion section. We also moved some of specifics about the results and parameter estimates to the Results section, where appropriate. We appreciate the reviewer’s recommendation and feel the paper is consequently much improved.

5) It would enhance the value of the paper if the authors more clearly highlighted the implications of the study findings (for theory, practice, and future research).

We have included implications in the revision of the Discussion section.

Reviewer 2 Report

Comments and Suggestions for Authors

Dear Authors,

Thank you for your important contribution. To enhance the clarity, impact, and scientific rigor of your manuscript, I would like to offer several general and specific suggestions before proposing updated literature for citation.

General Suggestions:

  • Clarify the hypothesis and objectives: The research questions or hypotheses could be stated more explicitly in the introduction to guide readers through the study’s purpose.

  • Strengthen the rationale: Provide a more detailed background on how your study fills gaps or addresses limitations in existing research.

  • Improve flow and coherence: Ensure smooth transitions between sections, especially when moving from molecular mechanisms to clinical implications, to maintain narrative continuity.

  • Highlight clinical relevance: Emphasize how your findings could impact diagnosis, treatment, or prognosis to appeal to a broader readership.

  • Data presentation: Where applicable, enhance figures and tables with clearer legends and ensure statistical analyses are fully described.

Specific Suggestions:

  • Expand the discussion on gene-environment interactions: Consider elaborating on how environmental factors like childhood adversity influence epigenetic changes linked to depression.

  • Include recent biomarker developments: Discuss novel biomarkers, including electrophysiologic coefficients, to provide insight into practical diagnostic tools.

  • Address comorbid conditions: Given the overlap between depression and eating disorders, it would be valuable to explore how these comorbidities affect quality of life and treatment approaches.

  • Discuss heterogeneity and personalized medicine: Incorporate perspectives on disease heterogeneity and the need for personalized treatment strategies based on biological and psychological profiling.

Proposed citations:

  1. Milic, J.; Jovic, S.; Sapic, R. Advancing Depression Management Through Biomarker Discovery with a Focus on Genetic and Epigenetic Aspects: A Comprehensive Study on Neurobiological, Neuroendocrine, Metabolic, and Inflammatory Pathways. Genes 2025, 16, 487. https://doi.org/10.3390/genes16050487
    To support and expand sections on genetic and epigenetic mechanisms and biomarker discovery.

  2. [Systematic review/meta-analysis on childhood adversity and epigenetics of depression] (https://pubmed.ncbi.nlm.nih.gov/36196689/)
    To deepen the discussion on environmental influences shaping depression risk via epigenetic changes.

  3. Milic, J., Stankic, D., Stefanovic, D. (2023). Eating Disorder and Quality of Life. In: Patel, V.B., Preedy, V.R. (eds) Eating Disorders. Springer, Cham. https://doi.org/10.1007/978-3-031-16691-4_21
    To contextualize comorbid eating disorders and their impact on patients' quality of life within the framework of mental health.

  4. Milic, J. (2022). How to design a reliable and practical biomarker: the electrophysiologic coefficient of depressiveness - δEPCD. Biomarkers, 27(8), 711–714. https://doi.org/10.1080/1354750X.2022.2122565
    To introduce innovative biomarker approaches that could improve clinical diagnostics.

  5. [Recent longitudinal study on biological and psychological mechanisms in depression] (https://www.sciencedirect.com/science/article/abs/pii/S0022395621002211)
    To underpin arguments about disease heterogeneity and the benefits of personalized treatment.

Comments on the Quality of English Language

Comments on the Quality of English Language

The manuscript is generally understandable; however, there are several areas where the clarity and fluency of the English language could be improved. Some sentences are overly complex or awkwardly structured, which occasionally hampers the flow and readability. Additionally, minor grammatical errors and inconsistent use of terminology appear throughout the text.

I recommend a thorough proofreading by a native English speaker or a professional language editing service to enhance the overall quality. Improving sentence structure, simplifying complex expressions, and ensuring consistent terminology will make the manuscript more accessible and impactful for a broader scientific audience.

Author Response

1) Clarify the hypotheses and objectives: The research questions or hypotheses could be stated more explicitly in the introduction to guide readers through the study’s purpose.

We originally described the hypotheses in the Estimate Model section; however, our description may not have been clear to readers. We have revised this section to articulate the hypotheses more clearly.

2) Strengthen the rationale: Provide a more detailed background on how your study fills gaps or addresses limitations in existing research.

In our revision of the Introduction section, we have attempted to clarify how our study makes a unique and important contribution to the literature. We hope the purpose of the study is now clearer to future readers.

3) Improve flow and coherence: Ensure smooth transitions between sections, especially when moving from molecular mechanisms to clinical implications, to maintain narrative continuity.

Our study did not involve molecular mechanism or a clinical perspective. Perhaps the reviewer’s comments refer to a different manuscript submitted for review.

4) Highlight clinical relevance: Emphasize how your findings could impact diagnosis, treatment, or prognosis to appeal to broader readership.

This study did not involve a clinical trial, clinical sample, or clinical intervention. It would be difficult to speculate about how the findings have clinical relevance. The findings have relevance for practitioners working with at-risk youth, especially justice-involved youth. We have revised the Discussion section to specify implications of the study.

5) Data presentation: Where applicable, enhance figures and tables with clearer legends and ensure statistical analyses are fully described.

The figure seems clearly and fully described by the description included with the title. The tables are also fully described. However, we added to the main header of Tables 2 and 3 that the estimated reported are the unstandardized estimates.

6) Expand the discussion on gene-environment interactions: Consider elaborating on how environmental factors like childhood adversity influence epigenetic changes linked to depression.

Gene-environment interactions are not relevant to the topic of our study. Perhaps the reviewer’s comments refer to a different manuscript submitted for review.

7) Include recent biomarker developments: Discuss novel biomarkers, including electrophysiologic coefficients to provide insight into practical diagnostic tools.

Biomarkers are not relevant to the topic of our study. Perhaps the reviewer’s comments refer to a different manuscript submitted for review.

8) Address comorbid conditions: Given the overlap between depression and eating disorders, it would be valuable to explore how these comorbidities affect quality of life and treatment approaches.

Our study did not examine eating disorders as a covariate of depression. Perhaps the reviewer’s comments refer to a different manuscript submitted for review.

9) Discuss heterogeneity and personalized medicine: Incorporate perspectives on disease heterogeneity and the need for personalized treatment strategies based on biological and psychological profiling.

A discussion of personalized medicine would not be appropriate in our manuscript. Perhaps the reviewer’s comments refer to a different manuscript submitted for review.

10) The manuscript is generally understandable; however, there are several areas where the clarity and fluency of the English language could be improved. Some sentences are overly complex or awkwardly structured, which occasionally hampers the flow and readability. Additionally, minor grammatical errors and inconsistent use of terminology appear throughout the text.

We have revised the manuscript to improve clarity, fluency, and grammar.

Reviewer 3 Report

Comments and Suggestions for Authors

I ask the authors to explain more precisely why (and which) ACEs increase the risk of depression. Also, I ask the authors to better specify why their sample results in a higher risk of depression. That is, why you chose to investigate this dimension in this specific population. Also, the authors should better elaborate on the instrument section and adequately report the statistical indicia of the scales used.

The authors should elaborate more clearly what is the novel element that the study introduces, because it is not immediately well expressed.

Clearly express the hypotheses by dedicating an ad hoc section!

The discussion is not a discussion.

macano practical implications, theoretical, limitations of the research, and more that could lead the read to a critical analysis of the work.

Author Response

1) I ask the authors to explain more precisely why (and which) ACEs increase the risk of depression.

We have revised the Introduction significantly. We added a new first paragraph that describes ways that ACEs and trauma affect brain function and behavior. We have also provided more information about how ACEs are associated with depression and differences in justice-involved youth, gender, and race/ethnicity. While there is no conclusive evidence of exactly how trauma shapes biological function and behavior, there is a clear association. The process is obviously complex and more research is needed.

2) Also, I ask the authors to better specify why their sample results in a higher risk of depression. That is, why you chose to investigate this dimension in this specific population.

We have provided more information about the prevalence of depression among justice-involved youth, which our sample reflects, and gender and race/ethnicity difference. We have also noted certain gaps in the literature that our study attempts to address.  

3) Also, the authors should better elaborate on the instrument section and adequately report the statistical indicia of the scales used.

We have added the Cronbach’s alpha score for the CSE-D items that comprised the latent scale. Thank you for reminding us to include this, as this will permit other scholars to compare the cohesion of the scale in this study with those reported elsewhere.

We also clarified that the ACEs items were dichotomous (0 = no, 1 = yes). The total ACEs score was used in the study. We did not include alpha scores for the ACEs index as it was comprised of 10 dichotomous variables. The literature on ACEs generally utilizes a total score for the dichotomous variables.

We also added content to the description of the sexual assault variable. We clarified that the correlation between the two measures was low (r = 0.28) and likely did not introduce multicollinearity in the tested multigroup models.

4) The authors should elaborate more clearly on what is the novel element that the study introduces, because it is not immediately well expressed.

In our revision of the Introduction section, we have attempted to clarify how our study makes a unique and important contribution to the literature. We hope the purpose of the study is now clearer to future readers.

5) Clearly express the hypotheses by dedicating an ad hoc section!

We originally described the hypotheses in the Estimate Model section; however, our description may not have been clear to readers. We have revised this section to articulate the hypotheses more clearly.

6) The discussion is not a discussion…practical implications, theoretical, limitations of the research, and more that could lead the read to a critical analysis of the work.

We have revised the Discussion section to elaborate on the findings and implications and highlight how the results compare to other studies. References are now included in the Discussion section. We also moved some of specifics about the results and parameter estimates to the Results section, where appropriate. We appreciate the reviewer’s recommendation and feel the paper is consequently much improved.

Round 2

Reviewer 1 Report

Comments and Suggestions for Authors

No comments

Author Response

Thank you.

Reviewer 3 Report

Comments and Suggestions for Authors

I am very satisfied with the work done by the authors. In my opinion, the paper has now improved and can be considered for publication. However, there are still some points that could be improved, and I expect the authors to be able to easily include the following requests:

- Describe the sociodemographic characteristics of the sample and indicate the nationality of the study in the abstract. 
- When indicating the possible consequences of ACE, take into account that you are dealing with adolescents and young adults. Therefore, indicate the main areas of functioning affected in adolescents, citing recent meta-analyses in particular. For example, the impact of ACE on body image disorders is significant and should be mentioned, listing the different outcomes (https://doi.org/10.1016/j.bodyim.2022.01.003).
- Create a section on the practical implications of your research.
- Create a section on the limitations of the research and offer clear directions for future research based on the limitations you have described. Try to expand on the limitations you have described, as the narrative is not complete.

Author Response

Reviewer comment #1:

Describe the sociodemographic characteristics of the sample and indicate the nationality of the study in the abstract.

Response: We have included text in the abstract to address this request.  

male (n=226) and female (n=98) youth entering a post-arrest intake facility in the Florida, U.S.A. juvenile justice system in 2024-2025. The youths averaged 15 years in age.

Reviewer comment #2:

When indicating the possible consequences of ACE, take into account that you are dealing with adolescents and young adults. Therefore, indicate the main areas of functioning affected in adolescents, citing recent meta-analyses in particular. For example, the impact of ACE on body image disorders is significant and should be mentioned, listing the different outcomes (https://doi.org/10.1016/j.bodyim.2022.01.003).

Response: We have added the following requested text to the introduction section of the manuscript and its associated reference:

Adverse childhood experiences are also risk factors for various mental health problems, including PTSD, anxiety, and body dysmorphic disorder[4].    

Reviewer comment #3:

Create a section on the practical implications of your research

Response: We have added a paragraph to the manuscript addressing this important point:

  1. Implications of our Research for Practice

The results of our research strongly recommend that youth entering the justice system be routinely screened for childhood adversity, abuse experiences, and mental health issues in a manner that elicits honest reporting of these experiences, with particular reference to gender and race/ethnicity differences in these phenomena. It is, of course, critical that these screenings be followed, where indicated, by full assessments and treatment follow-up. Unfortunately, such cascade of treatment services is not routinely available in many juvenile justice settings. Continued advocacy is needed to address this public health crisis.                                           

Reviewer comment #4:

Create a section on the limitations of the research and offer clear directions for future research based on the limitations you have described. Try to expand on the limitations you have described, as the narrative is not complete.

Response:

We have added a paragraph in the new study limitations section of the manuscript

  1. Study Limitations and their Implications

Future studies should explore the impact of positive versus adverse childhood experiences on mental health among justice-involved youth. In this vein, future research should follow a service system cascade model involving the identification of service need followed by indicated intervention/treatment engagement within a longitudinal framework to address the long term psychosocial and behavioral manifestations of ACES and trauma abuse experiences[27].